# Molecularly Imprinting Microfiltration Membranes Able to Absorb Diethyl Phthalate from Water

**DOI:** 10.3390/membranes12050503

**Published:** 2022-05-08

**Authors:** Katarzyna Smolinska-Kempisty, Joanna Wolska, Marek Bryjak

**Affiliations:** Department of Process Engineering and Technology of Polymer and Carbon Materials, Wroclaw University of Science and Technology, Wyb. St. Wyspianskiego 27, 50-370 Wroclaw, Poland; joanna.wolska@pwr.edu.pl (J.W.); marek.bryjak@pwr.edu.pl (M.B.)

**Keywords:** molecularly imprinting polymers, micropollutants, membrane filtration, phthalates

## Abstract

In this study, polypropylene porous membranes with an average pore size of 1.25 µm were modified by barrier discharge plasma. Next, molecularly imprinted layers with an imprint of diethyl phthalate (DEP) ware grafted of their surface. In order to optimize the composition of the modifying mixture various solvents, the ratios of functional monomers and the cross-linking monomer as well as various amounts of phthalate were verified. It was shown that the most effective membranes were obtained during polymerization in n-octane with the participation of functional monomers in the ratio 3:7 and the amount of phthalate 7 wt.%. The membranes were tested in the filtration process as well as static and dynamic sorption. In all of these processes, the imprinted membranes showed better properties than those without the imprint. The diethyl phthalate retention coefficient was 36.12% for membranes with a grafting yield of 1.916 mg/cm^2^. On the other hand, DEP static sorption for the imprinted membranes was 3.87 µmol/g higher than for non-imprinted membranes. Also, in the process of dynamic sorption higher values were observed for membranes with the imprint (DS_MIM_, 4.12 µmol/g; DS_NIM_, 1.18 µmol/g). The membranes were also tested under real conditions. In the process of filtration of tap water contaminated with phthalate, the presence of imprints in the membrane structure resulted in more than three times higher sorption values (3.09 µmol/g) than in the case of non-imprinted membranes (1.12 µmol/g).

## 1. Introduction

The dynamic development of technology offers many benefits but is connected with the generation of a large number of pollutants. Harmful substances enter the air, soil, and water [1,2,3]. Now, the plastic industry is accused of playing a great part in this pollution. [4]. The extensive development of polymer processing requires some additives with plasticizers that reduce intermolecular interactions and increase the mobility of polymer chains. As a result, the hardness of a material is reduced and its usability increases [5,6,7,8]. Finally, the demand for plasticizers in the plastics industry has grown enormously. In 2019, the global plasticizers manufacturing market was worth $93.76 billion and is estimated to reach $111.38 billion in 2023 [9].

One of the most popular groups of plasticizers are phthalates. They are characterized by their high availability and low price. At the same time, they belong to the group of endocrines disruptors, (EDs). Some phthalates can affect the endocrine systems, male fertility, and cause reproductive problems [10,11,12]. For this reason, applications of some phthalates are prohibited in some EU countries. Directives such as the EEC Directive 2005/84/EEC, (EC) 10/2011 ban the use of phthalates in products for children [9,13]. However, these compounds are still widely used in many other countries. Thus, their penetration into the environment is limited but still takes place.

Due to phthalate esters (PEs) widespread domestic and industrial applications, it is not surprising that the ubiquity of PEs contaminants is noted in different aquatic systems including marine water, sediments, rivers, lakes, and wetlands, landfill leachates, and fresh water, wastewater treatment plants, atmospheric aerosols, and rain water [14,15]. They get way to reach water systems through effluent direct discharges, urban and agricultural land runoff, leaching from waste dumps and other diffuse sources [14]. The reports available show that PEs occur in a wide concentration range from trace (μg/L) to mg/L depending on the degree of PEs application [14]. A large variety and abundance of PEs can be observed in aquatic systems in different countries [14,15,16].

A lot of publications describe methods for removing diethyl phthalate (DEP) from water. They include sorbents such as biochar [17], core–shell magnetic adsorbents [18], molecularly imprinted particles [19,20,21,22] or, for example, silver modified roasted date pits [23]. Some other advanced processes such as strong ionization discharge [24], dispersive liquid-liquid microextraction [25], semiconductor photocatalysis [26], non-thermal air plasma treatment [27] are also described. There are some reports about application of membrane processes for the removal of phthalates from water. They are as follows: reverse osmosis (RO) and nanofiltration (NF) [28], ultrafiltration (UF) with membrane coated by CuO/TiO_2_ [29] or, for example, membrane bioreactors (MBR) [30]. 

Currently, biodegradation under aerobic conditions seems to be the best option for the decomposition of phthalate esters in the environment. However, this is not an ideal solution, as the cut chains of phthalates are still present in the environment. Moreover, phthalate in wastewater may be removed by a membrane bioreactor treatment process or in a combined treatment process [31]. The combination of anaerobic wastewater treatment followed by a membrane bioreactor allowed obtaining a removal efficiency of about 65–71% and increased to 95–97% after MBR treatment. In a pilot anoxicoxic (AO) system, wastewater treatment followed by reverse osmosis and ultrafiltration, the phthalate ester removal efficiency increased from 75–80% for AO treatment to 93–97% with UF treatment [32]. However, the effectiveness of these processes depends on many factors and, in some areas, requires energy expenditure. The presented membrane allowed for achieving much lower phthalate retention rates, but it should be remembered that the tests were carried out for a single membrane. It would be necessary to continue them and check the effectiveness in the case of e.g., a pile of membranes or a capillary installation. Therefore, the efficiency of the presented membrane cannot be compared with the processes already carried out on an industrial scale.

The presented manuscript connected the phenomenon of molecular imprinting with membrane filtration in order to obtain an economical and effective membrane. The idea of molecular imprinting is based on the formation of a pre-polymerization complex of the template with monomers. During polymerization, the template is encased in a polymer network. After removing the template, an imprint of its shape remains in the polymer matrix [33,34,35,36]. Currently, the molecularly imprinted membranes could be used as sensors, for sewage treatment, in food science, synthesis of biomacromolecules, drug delivery systems, separation of herbal compounds, and many more [37,38,39,40]. The most popular method of preparing the imprinted membranes is sol-gel precipitation [41,42,43]. However, other methods, such as the phase inversion technique and the surface imprinting via graft copolymerization, are often used [39,40,44,45,46].

The main aim of the presented paper was to obtain molecularly imprinted microfiltration membranes sensitive for diethyl phthalate. The method of surface activation by plasma discharge and grafting the layer of molecularly imprinted. Plasma treatment is ecofriendly and fast method of surface activation. 

To the best of our knowledge, this is the first approach for the preparation of DEP-sensitive membranes by grafting from the method induced by plasma treatment. The principle of operation of the molecularly imprinted membrane, (MIM), is schematically presented in Figure 1.

## 2. Materials and Methods

### 2.1. Materials

Porous membranes made from polypropylene (PP) (1.25 µm, Millipore Poznan, Poland) were used for modification. Before modification, the membranes were rinsed with ethanol and dried at room temperature. All solvents (toluene, cyclohexanol, chloroform, n-octane), N-isopropylacrylamide (NIPAM), ethylene glycol dimethacrylate (EGDMA), methyl methacrylate (MMA), azobisisobutyronitrile (AIBN), diethyl phthalate (DEP), dibutyl phthalate (DBP) were purchased from Sigma-Aldrich (Poznan, Poland) and used without additional purification.

### 2.2. Modification Process

#### 2.2.1. Plasma Treatment

For surface modification of polypropylene membranes, the dielectric barrier discharge plasma device (Dora Power System Ltd., Wroclaw, Poland) was used. The following plasma parameters were used: voltage 20 kV, current 5 mA, argon flow 30 L/h, modification time 30 s (each side of the filter). After plasma activation, the membranes were kept 5 min in air.

#### 2.2.2. Grafting Procedure

Membranes, after plasma modification, were immersed in a 50% solution of monomers in an organic solvent. The composition of the polymerization mixtures is presented in Table 1. The 2 wt.% of AIBN according to the number of monomers was added to the polymerization mixtures. As a cross-linker, ethylene glycol dimethacrylate and, as functional monomers, methyl methacrylate and N-isopropylacrylamide were used. During the preparation of the polymerization mixture, ratios of monomers and a cross-linker were calculated. The amount of DEP was 0, 5, 7, and 10 wt.% with respect to the monomers. The samples were immersed in a given polymerization mixture for 24 h at ambient temperature. After this time, they were dried between filtration papers to remove excess monomers. Then, they were placed between the foil sheets and polymerized. Polymerization was carried out for 24 h at 60 °C in the oven. After modification, the non-polymerized components were extracted in Soxhlet with ethanol for 12 h. In the case of absence of DEP in the polymerization mixture, the non-imprinted membranes (NIMs) were obtained.

### 2.3. Membrane Characterization 

#### 2.3.1. Grafting Yield

The grafting yield, (*GY*) (mg/cm^2^), was calculated according to Equation (1):(1)$$GY=\left(\frac{M_{1}-M_{2}}{M_{1}}\right)/A$$
where: *M_1_* and *M_2_* are the weights of the membrane after and before modification (g) and *A* is the surface area of the membrane (cm^2^) (A = 17.3 cm^2^).

#### 2.3.2. Water Flux

Membranes were hydrophilized by immersion in 96% ethanol prior to measurement. Water flux, (*J*) (L/m^2^ h), through the membranes was measured in the Amicon 8050 cell (Poznan, Poland) with a pressure of 0.002 MPa and calculated according to Equation (2): (2)$$J=\frac{V}{At}$$
where: *V* is the volume of permeate (L), *A* is the surface area of the membrane (m^2^), and *t* is the time measured for 25 mL of permeate (h).

#### 2.3.3. Average Pores Size

The pores size, (*r*) (µm), was calculated according to Equation (3):(3)$$r=\sqrt{\frac{Jd\eta }{p\varepsilon }}$$
where: *J* is the water flux, *d* is the membrane thickness, *η* is the water viscosity, *p* is pressure, and *ε* is porosity.

#### 2.3.4. Rejection Coefficient 

The rejection coefficient, (*R*) (%) was calculated according to Equation (4) after filtering 10 mL of 30 mg/L of aqueous DEP solution at the gravity pressure of the liquid column.
(4)$$R=\left(1-\frac{C_{p}}{C_{0}}\right)100$$
where *C_0_* is the initial concentration of DEP (mg/L), and *C_p_* is the concentration of DEP in permeate (mg/L).

#### 2.3.5. Static Sorption

DEP sorption was carried out at 30 mg/L in aqueous solutions at ambient temperature. Membranes were shaken with 20 mL of solution for 24 h. After equilibrium was reached, the membranes were separated from the solution. The supernatant was analyzed to determine the concentration of DEP that remained in the solution using the Jasco V-630 UV–VIS spectrophotometer (Medson, Paczkowo, Poland) at 229 nm. The concentration was calculated using the measured absorbance and was based on the previously prepared DEP calibration curve in deionized water. The static sorption, (*SS*) (mg/g), was calculated according to Equation (5).
(5)$$SS=\frac{\left(C_{1}-C_{0}\right)V}{M_{2}-M_{1}}$$
where: *C*_0_, *C*_1_ is the DEP concentration before and after sorption, respectively (mg/L), *M*_1_ and *M*_2_ are the membrane weights after and before modification, respectively (g), and *V* is the volume of the DEP solution (L).

#### 2.3.6. Dynamic Sorption

Dynamic sorption properties were evaluated by filtrating 25 mL of 30 mg/L aqueous solution of DEP (Amicon 8050, cell filled with 50 mL of feed, 0,002 MPa). Phthalate concentration was measured in permeate. The sorption (*DS*) (mg/g), was calculated according to Equation (6):(6)$$DS=\frac{\left(C_{p}-C_{f}\right)V_{p}}{M_{2}-M_{1}}$$
where *C_p_* and *C_f_* are the concentration of DEP in the permeate and feed, respectively (mg/L), *V_p_* is the volume of the permeate (L), and *M*_1_ and *M*_2_ are the weights of the membrane after and before modification, respectively (g).

#### 2.3.7. Sorption Kinetics 

For the sorption kinetics, MIM and NIM membrane samples (G-series, GY = 1.16 mg/cm^2^) were placed in the Amicon cell and solution of DEP in the deionized water was filtrated through the given membrane under 0.002 MPa at room temperature. The 5 mL of filtrate was collected during the filtration and analyzed by determining the concentration of DEP in the permeate using the Jasco V-630 UV-VIS spectrophotometer. After this, the sorption was calculated. The process of sorption was carried out until the concentration of DEP in the filtrate was the same as in the feed solution.

#### 2.3.8. Sorption Isotherms

The sorption isotherms were determined by contacting a given membrane MIM or NIM (G-series) with 20 mL of solution containing 30 mg/L of DEP. The samples were shaken for 24 h at room temperature. After this time, the membranes were removed from the solution and the concentration of diethyl phthalate was determined by UV-VIS (λ = 229 nm).

#### 2.3.9. Dynamic Sorption in Real Samples

The 30 mg/L DEP solution in tap water was used as a real sample. The sorption parameter was evaluated by filtering 10 mL in the Amicon 8050 cell, under the pressure of 0.002 MPa.

#### 2.3.10. Selectivity of Membranes

Samples of selected membranes (G-series), with and without imprints and with the grafting yield 1.32–1.44 mg/cm^2^, were shaken with 20 mL of solutions of two phthalates: diethyl phthalate and phthalate dibutyl, at ambient temperature for 24 h. The initial concentration of each phthalate was 0.005 mmol/L. After the process of static sorption, the membranes were removed from the solution, and the concentrations of DEP and DBP were investigated with the Jasco UV-VIS V = 630 spectrophotometer at 229 nm and 274 nm, respectively. The static sorption (*SS*) of each phthalate for both samples were calculated from Equation (5). This parameter allowed us to determine the distribution coefficients (*K*), defined as the amount of a substance adsorbed by defined unit of weight of an adsorbent to the amount of the substance in the same volume of the solution after sorption. It was calculated using Equation (7) [20,47]:(7)$$K=\frac{SS_{e}q}{C_{e}}$$
where *SS_e_* is amount of phthalate adsorbed at equilibrium (mmol/g), *C_e_* is liquid-phase concentrations of analyte at equilibrium (mmol/L), *q* is the density of solution (g/L). 

Next the sorption selectivity of one substance versus another was determined by selectivity coefficient, (α) using Equation (8) [47], which is the ratio of the distribution coefficients of the two compounds (in this case, DEP and DBP).
(8)$$\alpha =\frac{K_{DEP}}{K_{DBP}}$$where *K_DEP_* is a distribution coefficient of diethyl phthalate and *K_DBP_* is a distribution coefficient of dibutyl phthalate.

In the next step, the imprinting factor (*IF*) was calculated as [20,48,49,50]:(9)$$\mathrm{IF}=\frac{K_{i\left(j\right)}^{\mathrm{MIP}}}{K_{i\left(j\right)}^{\mathrm{NIP}}}$$
where *K_i(j)_* is the distribution coefficient (*i* = DEP, *j*= DBP)) of a phthalate on an imprinted sample (*K^MIP^*) and on a non-imprinted sample (*K^NIP^*). To obtain the specific selectivity factor (*S*), the ratio of *IFs* for two different substances (*IF_i_*, *IF_j_*) was calculated using Equation (10):(10)$$S=\frac{{\mathrm{IF}}_{i}}{{\mathrm{IF}}_{j}}$$

#### 2.3.11. Membrane Stability in Sorption/Desorption Cycles

The selected samples of MIM and NIM membranes with the grafting yield 1.09 mg/cm^2^ were tested in the cycles of sorption and desorption the process of static sorption of selected materials repeating the procedure described about three more times. For the static sorption tests, the DEP model solution was used with the concentration of 30 mg/L. After each cycle, the static sorption capacity was evaluated. To make the comparison easy, sorption was normalized to the fresh membrane. The uptake of DEP was calculated from Equation (5) and the percentage of initial DEP removal (*%_SSDEP_*) was calculated from Equation (11).
(11)$${\%}_{\mathrm{SSDEP}}=\frac{{\mathrm{SS}}_{n}}{{\mathrm{SS}}_{0}}100\%$$
where *SS*_0_ is the static sorption of DEP on the fresh membrane (MIM or NIM) (mg/g), *SS_n_* is the static sorption on the regenerated membrane (MIM or NIM) (mg/g), *n* is the cycle of sorption/desorption process.

During the desorption process, membranes were regenerated in the Soxhlet with the use of methanol for 1 h and again process of sorption was performed.

#### 2.3.12. Contact Angle and Surface Energy

The dynamic contact angles of water were measured using the PG-X goniometer, Fibro Systems (Wroclaw, Poland). Measurements were carried out on both non- and modified membranes. The total surface energy of the membrane was calculated according to the ASTM D5946 PG norm.

#### 2.3.13. IR Spectroscopy

To characterize the membranes obtained, the middle infrared spectra (4000–400 cm^−1^) were collected by means of a Fourier-transform, Jasco FT/IR-4700 (Medson, Paczkowo, Poland) model spectrometer. MIM or NIM was placed on the diamond crystal of the attenuated total reflectance device. The spectral data were recorded at a resolution of 4 cm^−1^ by collecting 64 scans. The collected data were elaborated using Jasco Spectra Manager software.

#### 2.3.14. Scanning Electron Microscopy

The morphology of the prefilter surface was investigated by a Carl Zeiss EVO LS15 SEM microscope (Krakow, Poland), operated at 5.00 kV. Prior to the microscopy evaluation, the membrane surface was sputter coated with gold.

## 3. Results

A molecularly imprinted polymer layer has been applied on porous polypropylene membranes. Plasma dielectric barrier discharges were used to modify the surface of the membranes. In effect, peroxide and/or hydroperoxide functionality have been generated which created specific places for fixing the applied layer with an imprint on the membrane. The sorption character of the process was verified by performing static and dynamic sorption. Static sorption was carried out for 24 h, while dynamic sorption was performed in a continuous flow in Amicon 8050. 

In order to obtain sensitive MIMs, different compositions of the polymerization mixture were tested. First, the influence of DEP concentration in the polymerization mixture was tested. The amount of the template is related to the number of active imprints formed on the membrane surface layer. Due to the complicated DEP imprinting process, it was difficult to suggest the best amount of the template. It is known that higher concentration results in more imprints but it can cause imprints overlapping and reduction of the membrane affinity.

In addition, various solvents were used to prepare the polymerization mixture. The goal was to select an appropriate porous agent and increase the effectiveness of the prepared membranes. The more porous structure facilitates access to the active sites. The results are shown in Table 2. Toluene (series A), cyclohexanol (series B), chloroform (series C), n-octane (series D and E) and the n-octane/toluene mixture 1:1 (series F) were used. In the case of the A-C and F series, the sorption values for MIP and NIP were very similar to each other. Sorption increases with increasing amount of the applied layer, but it was nonspecific. This allows us to conclude that in the selected polymerization mixture, DEP did not imprint into the polymerized layer or the number of imprints is too small to notice the differences between MIM and NIM. Some differences between MIM and NIM have been noted for grafting from n-octane (MIM 4.23 mg/g–NIM 3.08 mg/g). For this reason, the amount of DEP in the polymerization mixture was increased from 5 wt.% to 7 wt.%. When the template concentration was increased, the difference between MIM and NIM increased, as well as the sorption value, to 5.7 mg/g. 

In the next step, the ratio of functional monomers was changed (i.e, MMA to NIPAM) three to seven and four to six respectively (Table 3).

Grafting from a mixture in which the ratio of MMA to NIPAM was 3:7 allowed to obtain imprinted membranes that were able to absorb almost four times more DEP than membranes without imprint. However, changing the ratio of functional monomers to 4:6 and increasing the amount of DEP in this case to 10 wt.% did not increase the sorption or the difference between MIM and NIM

The molecular imprinting process, although discovered many years ago, is still in the development phase. This is most likely due to the fact that each template and monomer system requires individual consideration. Even slight changes in the composition of the polymerization mixture can affect the efficiency of the imprinting process. In our previous work [16,17,18,19,28,29,42,50,51], we presented materials with an imprint for compounds belonging to the plasticizer group and registered as endocrine disruptors. Studies with templates such as diethyl phthalate [17,29] or bisphenol A [19,28,42] were carried out. In all of them, the main functional monomer was methyl methacrylate, and an additional monomer (N-isopropylacrylamide, acrylic acid) was copolymerized. The solvent was n-octane or toluene. In the case of the analyzed systems, sorption values for MIP were obtained higher than those for NIP. Most probably, this was due to the structure of the materials obtained. Core-shell layers [50] and imprinted membranes [52] were studied for the removal of BPA. In these cases, the best solvent was n-octane. These articles confirmed that the optimization of the grafting parameters should be performed separately for each system.

In the presented work, the use of toluene as the porous agent allowed to obtain higher sorption values for imprinted membranes (5.70 µmol/g) than for non-imprinted analogues (3.89 µmol/g). When n-octane was used as porogen, the obtained MIM sorption value was the same as that of toluene. However, the sorption value for NIM was lower (1.86 µmol/g). It was probably caused by the presence of an aromatic ring in toluene, which left an imprint in the polymer matrix. In the sorption process, phthalate had affinity to imprints formed by toluene. The fact that n-octane seemed to be better than toluene can also be explained by the Hilderbrand solubility parameter (HSP). MMA and n-octane show a HSP difference of 1.1 units and DEP and n-octane of 2.0 units. This allows us to conclude that MMA could form a stable complex with DEP. In the case of the other solvents, these proportions were opposite or close to each other [52,53]. On the other hand, the chemical structure of the compounds used should also be considered. DEP has an aromatic-aliphatic structure. Perhaps for this reason, aliphatic n-octane turned out to be a better solvent than aromatic toluene for the layer grafted on the PP membrane. In summary, molecular imprinting is a very advanced process that is related to many factors. For this reason, it is very difficult to compare the sorption properties of the obtained materials with those of some published papers. Previously, we presented core-shell layers with BPA imprints with sorption properties at a level of 0.09 mmol/g [50]. In the case of MIP layered membranes with BPA imprints, the sorption reached 0.06 mmol BPA/g [51]. It should be noted that thin layers grafted on the membrane surface show different sorption affinity that core-shell particles.

In the next step, the synthetized materials were tested in real conditions. G-series membranes were used in the filtration process of the DEP contaminated model solution and tap water. It was calculated that during filtration they sorbed almost three times more phthalate than membranes without imprint (Table 4).

The selected materials were then used in the dynamic sorption process. This allows us to observe the sorption kinetic. The relationship between dynamic sorption (*DS*) and contact time is shown in Figure 2. It can be concluded that DEP sorption during filtration has more effectiveness on the MIP membrane than on the non-imprinted sample. Diethyl phthalate was quickly sorbed in the first minutes of the process (50% of maximum uptake was observed after about 1.5 min for MIM and 2 min for the NIM membrane). 

In order to predict the mechanism involved in the sorption process, the Lagergren pseudo-first- and pseudo-second-order kinetic models were applied [54,55].

The sorption kinetics following the pseudo-first-order model is given by Equation (12) [55,56]:(12)$$\frac{\mathrm{dDS}}{\mathrm{dt}}=k_{1}\left({\mathrm{DS}}_{e}-{\mathrm{DS}}_{t}\right)$$
where *DS_t_* and *DS_e_* represent the number of adsorbed species (mg/g) at any time *t* and at equilibrium time, respectively, and *k*_1_ represents the sorption rate constant (1/min).

Integrating Equation (13) with respect to the boundary conditions *DS* = 0 at *t* = 0, and *DS* = *DS_t_* at *t* = *t*, one obtains:(13)$$\log\left({\mathrm{DS}}_{e}-{\mathrm{DS}}_{t}\right)=\log\left({\mathrm{DS}}_{e}\right)-\frac{k_{1}t}{2.303}$$

Sorption rate constant *k*_1_ (1/min) can be calculated from the plot of *log (DS_e_−DS_t_)* versus time. 

The kinetic data can also be analyzed by means of pseudo-second-order kinetics. According to this model, the sorption behavior is controlled by the chemisorption process occurring either electronic sharing or electronic exchange. This model is represented by Equation (14) [54]:(14)$$\frac{\mathrm{dDS}}{\mathrm{dt}}=k_{2}{\left({\mathrm{DS}}_{e}-{\mathrm{DS}}_{t}\right)}^{2}$$
where *k*_2_ is the pseudo-second-order rate constant (1/min), *DS_e_* and *DS_t_* are the number of adsorbed species (mg/g) at equilibrium and at time *t*, respectively. Varying the variables in Equation (15) one gets:(15)$$\frac{\mathrm{dDS}}{\left({\mathrm{DS}}_{e}-{\mathrm{DS}}_{t}\right)}=k_{2}\mathrm{dt}$$
and integrating Equation (16) for the boundary conditions *DS* = 0 at *t* = 0 and *DS* = *DS_t_* at *t* = *t*, one obtains the final form:(16)$$\frac{t}{{\mathrm{DS}}_{e}}=\frac{1}{k_{2}{\mathrm{DS}}_{e}^{2}}+\frac{1}{{\mathrm{DS}}_{e}}t$$

A plot *t/DS* versus *t* gives the value of the constants *k*_2_. 

The experimental data were fitted to these two models that served to calculate the *k*_1_ and *k*_2_ constants. The results of the kinetic analysis are given in Table 5.

It can be seen that sorption kinetics for both materials fits well to the pseudo-second-order mechanism. During the evaluation of the kinetic data, the initial sorption data were calculated. The initial sorption rate was about seven times higher for MIM than for NIM. It means that the sorption on the MIP membrane is faster than that on the non-imprint analogues.

To get a better insight into the behavior of the binding sites, the study of the sorption on the prepared membrane in the presence of a mixture of two commonly existed in real samples phthalates, such as diethyl phthalate and dibutyl phthalate, was carried out. From the analysis of obtained data, the distribution coefficients (*K*) were calculated for both of phthalate [20,48]. To measure the imprinting effect, the partitioning effects encountered for each compound should be normalized. Therefore, the comparison of distribution coefficients for binding the same substance to MIM and NIM and determination of *IF* was carried out. Moreover, taking the ratio of *IF* for two substances, and therefore, eliminating partitioning effects between two molecules due to non-specific effects, a specific selectivity factor (*S*) was obtained [20,49]. The results are presented in Table 6. The higher DEP affinity to MIP and the effectiveness of molecular imprinting were proven by the high value of *IF* (2.5). Moreover, the selectivity coefficient (α) determined toward DBP for the MIM sample exceeds value 1 and is approximately two time higher than for NIM. This confirms that binding of DEP from the mixture of these two phthalates is favored. It can be ascribed to the presence of cavities created by imprinted, DEP inducing the effect of molecular recognition.

During evaluating the sorption properties of the prepared membranes, the sorption isotherms at room temperature were determined. The sorption isotherms for MIM and NIM materials are shown in Figure 3. As can be observed, the capacity of MIM is much larger than for NIM. For MIM, the sorption capacity toward diethyl phthalate was about 0.014 mg DEP/g while for NIM it was approximately 10 times lower and reached 0.004 mg DEP/g. Additionally, during comparison of the concentrations when maximum sorption capacities were achieved, it can be seen that for MIM, the maximum capacity is reached at lower concentrations than for NIM case.

To describe the interaction of diethyl phthalate with the prepared MIM and NIM, three adsorption isotherm models, Langmuir, Freundlich, and Dubinin–Radushkevich, were used. The Langmuir isotherm assumes the formation of a monolayer of molecules at the surface of the adsorbent. The linear form of the Langmuir equation is as follows Equation (17):(17)$$\frac{1}{{\mathrm{SS}}_{e}}=\frac{1}{{\mathrm{qSS}}_{m}b_{L}C_{e}}+\frac{1}{{\mathrm{SS}}_{m}}$$
where *SS_e_* is the static uptake at equilibrium concentration (mg/g), *SS_m_* is the maximal static uptake (mg/g), *C_e_* is the DEP equilibrium concentration (mg/L), and *b_L_* is the constant related to the binding energy of the sorption system (L/mg). Parameters *SS_m_* and *b_L_* were calculated from the slope and intercept of the linear plot of *1/q_e_* vs. 1/*C_e_* [57].

Additionally, a separation factor (dimensionless) called the equilibrium or separation parameter (*R_L_*) is determined during the analysis, which allows one to determine whether the Langmuir isotherm model is favorable for a given separation process Equation (18) [57]: (18)$$R_{L}=\frac{1}{1+b_{L}C_{0}}$$
where C_0_ is the initial adsorbate concentration (mg/L). 

The parameter *R_L_* indicates the efficiency of the adsorption process. The isotherm is (I) unfavorable when *R_L_* > 1, (II) linear when *R_L_* = 1, (III) favorable when *R_L_* < 1, and (IV) irreversible when *R_L_* = 0. Unfortunately, the obtained fitting data did not allow us to calculate the characteristic parameters for this isotherm model, which also confirms that the model is not suitable for the analysis of the sorption process of diethyl phthalate on all membranes studied [57].

The next investigated model was the Freundlich isotherm. This model is assumed as a power function relationship between *SS_e_* and *C_e_* and is easily applicable when the experimental data are plotted in *log q_e_* versus *log C_e_* format Equation (19) [54]. The Freundlich isotherm is applicable to adsorption processes that occur on heterogonous surfaces. This isotherm gives an expression which defines the surface heterogeneity and the exponential distribution of active sites and their energies. The linear form of the Freundlich isotherm is as follows [22,54,57]:(19)$${\mathrm{logSS}}_{e}=\frac{1}{n}{\;\mathrm{logC}}_{e}+\mathrm{loga}$$

In this model, there are two fitting parameters *a* and 1/*n* that both yield a measure of physical binding. The *a* parameter is the constant related to the adsorption capacity. The 1/*n* parameter is known as the heterogeneity index. For homogeneous materials, 1/*n* would be equal to 1, when the adsorption is linear, adsorption sites are homogenous in energy, and no interactions occur between the adsorbed compounds. On the other hand, when values of 1/*n* parameter approach to zero increase the heterogeneous character of the polymer. The constant *n* should have a value in the range of 1–10 for adsorption to be classified as favorable [22,57,58]. Unfortunately, the obtained fitting data did not allow to calculate the characteristic parameters for this isotherm model also, which confirms that this model is not suitable for the analysis of the sorption process of DEP on all studied membranes. The values R^2^ for the Langmuir and Freundlich models are given in Table 7 and they confirm that both of these models do not describe well the interactions between DEP molecules and membranes studied. 

The third investigated sorption model is the Dubinin–Radushkevich isotherm, which helps to study the interaction between the sorbate and the sorbent [54]. This approach is used generally to distinguish the kind of dominated sorption: physical or chemical one. The isotherm is expressed by Equation (20).
(20)$$\ln q={\mathrm{lnq}}_{m}-K_{\mathrm{DR}}\varepsilon^{2}$$
where *SS_m_* is the maximum adsorption capacity of the material (mmol/g), *K_DR_* is the Dubinin–Radushkevich constant (kJ^2^/mol^2^), *ε* is the Polanyi potential Equation (21):(21)$$\varepsilon =\mathrm{RT}\ln(1+\frac{1}{c})$$

*K_DR_* is related to the free energy (*E*, (kJ/mol)) of adsorption per molecule of adsorbate when it is transferred to the surface of the solid from infinity (in the solution). The adsorption behavior could predict physical adsorption in the range of 1–8 kJ/mol, and chemical adsorption at over 8 kJ/mol. The free energy can be calculated using Equation (22) [54].
(22)$$E={\left(2K_{\mathrm{DR}}\right)}^{-0.5}$$

During the analysis, it can be observed that this model is suitable only for the MIM material: the value of R^2^ is very close to 1 for this sample. In the case of NIM, the R^2^ value was lower than 1 (see Table 7). For NIM, it was not possible to calculate the characteristic parameters for this model. The free energy (*E*) value for MIM reached the value of 0.07 kJ/mol, much lower than 8 kJ/mol the critical value of chemical sorption. This means physical adsorption is dominated for DEP interactions with MIM.

To complete the characterization of the obtained materials, the cycles of sorption and desorption were performed. Such a procedure exposes materials (MIM and NIM) to different conditions and may affect their sorption properties. In order to check the membrane stability, three sorption/desorption cycles were conducted for both types of membranes (MIM and NIM). After each cycle, the static sorption capacity was evaluated. To make the comparison easy, sorption was normalized to the fresh membrane. The results are given in Figure 4. 

The data collected show that changes of conditions and acting during cycles of sorption and desorption did not deteriorate the sorption properties of the investigated sample. The sorption capacity for both MIM and NIM was almost the same as for fresh membranes (95–97%). During desorption, all sorbed DEP was desorbed. This phenomenon was also observed during another study in our research team, with the core-shell materials type [50].

Confirmation of the effectiveness of the modification can also be found in the analysis of the mass of the membranes before and after applying the imprinted layer. The weight of all samples increased after the grafting process. The water flux of the tested filters was also measured. On the basis of this, it was calculated how the average pore size of the membrane changed depending on the amount of the applied layer. 

Before modification, the membranes were characterized by an average water flux of approximately 120 L/m^2^ h (Table 6). As a rule, plasma modification causes etching of the modified materials. Thus, the hydraulic water flux after plasma modification may also increase. This phenomenon was observed in modified samples. As a result of the etching of the material surface, the flux increased by approximately 9 L/m^2^ h. The average pore size also increased by about 0.01 µm. This may be confirmation of the fact that the plasma modified not only the sample surface but also the interior of the membrane pores (each membrane was modified on both sides) [59,60].

Grafted on the membrane (G-series), a small amount of the imprinted layer about 0.026 mg/cm^2^ resulted in a minimal change in water flux to a value of 118 L/m^2^h. However, it did not affect the average pore size (Table 8). The experimentally determined porosity of the membranes before modification was between 27 and 45%. After modification, depending on the degree of grafting, it varied between 19 and 38%. The thickness of the membranes also changed from 150 µm before the modification to even 200 µm after the modification for the membrane with a mass increase of approx. 3.7 mg /cm^2^.

The reduction in the pore size began to be observed from the amount of about 0.5 mg/cm^2^ of the applied layer. This amount of material introduced caused the reduction of the flow through the membrane approximately 22 L/m^2^ h. The average pore size also decreased to about 1.18 µm. Subsequent modifications that focused on increasing the weight of the grafted layer caused a reduction in the water flux and the average pore size at the same time. Increasing the weight of the membrane by about 20% (1.9 mg/cm^2^) resulted in a reduction of the flux of about 50% compared to that before modification. The average pore size also decreased by approximately 0.36 µm (Table 8).

In order to verify the separation properties of the membranes, the rejection coefficient was also calculated. The average pore size of the modified membranes, even those containing quite large amounts of the grafted imprint layer, was significantly larger than the DEP molecules. Therefore, it is difficult to consider the sieve separation mechanism. However, the presence of the DEP imprint on the surface of the membranes and in the pores allowed for the retention of about 36% of the phthalate molecules present in the water. At the character of same time, the material filtering has been retained (Table 8, Figure 5). In our opinion, this represents a great advantage of these membranes for many potential applications. One of these could be the potential use of this type of membranes in the microfiltration process. In addition, to the traditional separation of phases, it would be possible to significantly reduce the amount of such a dangerous substance as DEP in filtered solution. These membranes could also be used for water polishing without any additional energy costs, since with such a large flux, they would not require the use of additional high pressures.

The membranes with the best sorption properties were characterized by the IR spectrum, SEM photos, and water contact angle of the surface measurements.

These analyzes were performed for the G series membranes. Figure 6 shows the spectrum of the membrane before (PP) and after modification (MIM-PP). In the case of modified membrane, it was possible to observe absorption bands, which were absent in the case of the membrane before the modification process. The absorption band at a wavenumber of about 1720 cm^−1^ is characteristic for the C=O group. This bond is found in the molecules of both NIPAM, MMA, and EGDMA. A slight band at 1536 cm^−1^ confirms the presence of the NH group derived from the PNIPAM chains present in the structure of the modified membrane. In addition, at a wavenumber of about 1132 cm^−1^, a quite wide band appeared ascribed to the C–O bond. This bond is present in the MMA and EGDMA molecule (Figure 6).

Figure 7 shows SEM pictures of membranes before and after grafting. Non-modified, polypropylene fibers had a smooth surface (A). However, after modification, changes on the surface were clearly visible, unevenness appeared, and additional thickenings. Some parts of the membrane seem to be completely covered with a layer of grafted material (B).

In order to verify the changes of a hydrophilic-hydrophobic nature that occurred on the surface of the membrane as a result of its modification, the dynamic contact angle of the material by water was measured. The results are presented in Table 9. They show that even a small amount of the grafted layer (0.046 g/cm^2^) allows one quite significantly increase the hydrophilicity of the surface, the contact angle decreases by about 21° and the surface energy increases to approximately 40.6 mJ/m^2^. Increasing the amount of imprinted layer to about 1 mg/cm^2^ increased the surface energy to a level of 52.2 mJ/m^2^.

## 4. Conclusions

This paper presents the possibility of obtaining DEP-sensitive filters that can reduce the content of DEP in the filtrate. Thanks to this, it is possible to reduce the risk of human exposure to this endocrine disrupting substance. However, it was necessary to select the polymerization parameters which allowed for obtaining effective membranes. The most suitable of the solvents tested for imprinting DEP on PP membranes seems to be n-octane. Effectively sorbing membranes were obtained in the grafting process of imprinted polymeric layers from a monomer solution with a ratio of MMA to NIPAM of 3:7. The amount of template was 7 wt.% with respect to the functional monomers. The membranes prepared in this way were tested in the process of filtration and static and dynamic sorption. They were characterized by a 36% retention coefficient despite the fact that they had a larger average pore size than the size of molecules of the removed DEP. In the process of static sorption from the model solution, the sorption of the imprinted membranes was almost four times higher than that on the filters without the imprint. The membranes were also tested under real conditions. Also, in this case, MIMs showed by 64% higher phthalate sorbing capacity than the analogous NIM. 

## Figures and Tables

**Figure 1 membranes-12-00503-f001:**
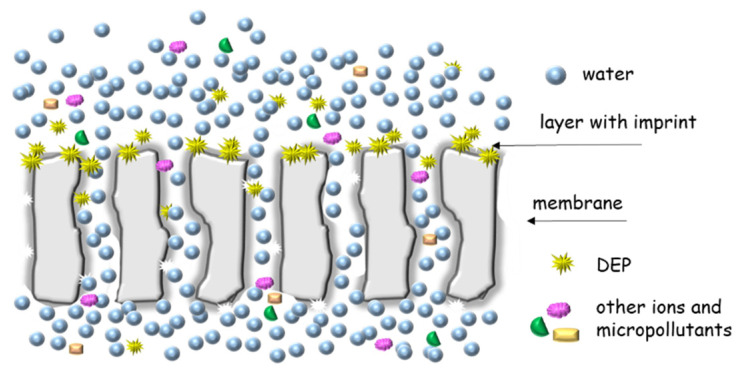
Scheme of molecularly imprinted membrane (MIM) operation.

**Figure 2 membranes-12-00503-f002:**
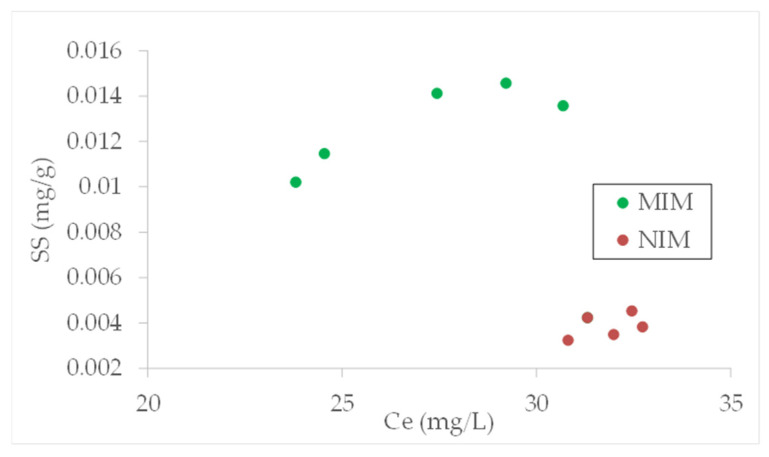
The relationship between dynamic sorption (DS) and contact time.

**Figure 3 membranes-12-00503-f003:**
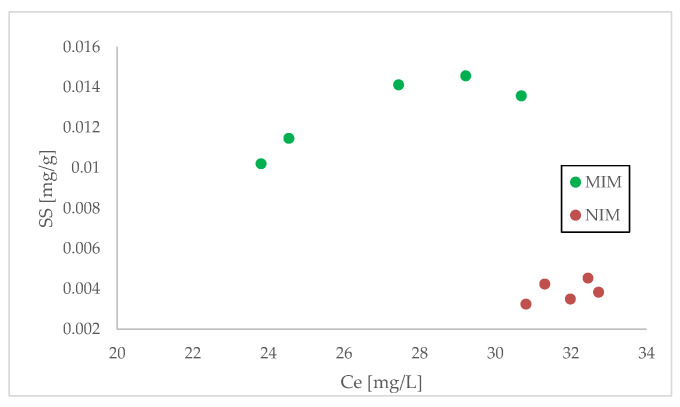
DEP sorption isotherms for resins molecularly imprinted membrane (MIM) and non-imprinted membrane (NIM) from series G.

**Figure 4 membranes-12-00503-f004:**
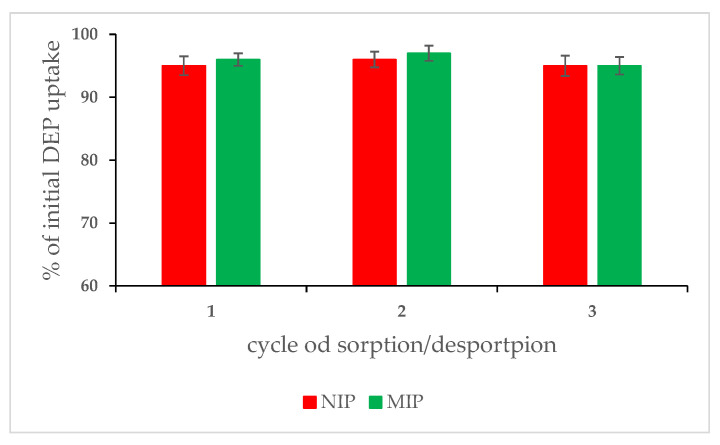
Sorption cycles of MIM and NIM in three cycles of sorption and desorption.

**Figure 5 membranes-12-00503-f005:**
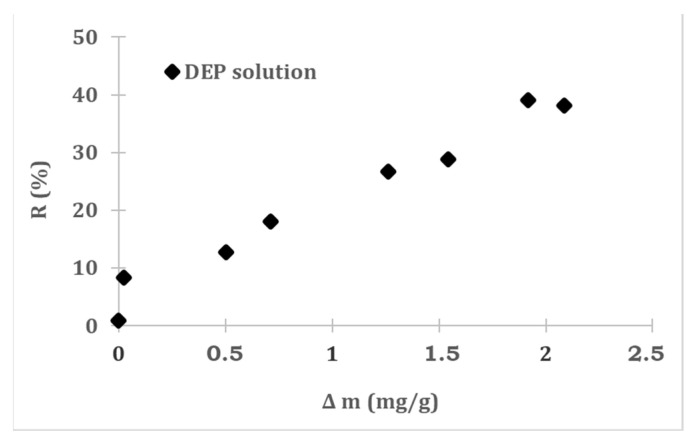
Dependence of the rejection coefficient on the amount of grafted layer.

**Figure 6 membranes-12-00503-f006:**
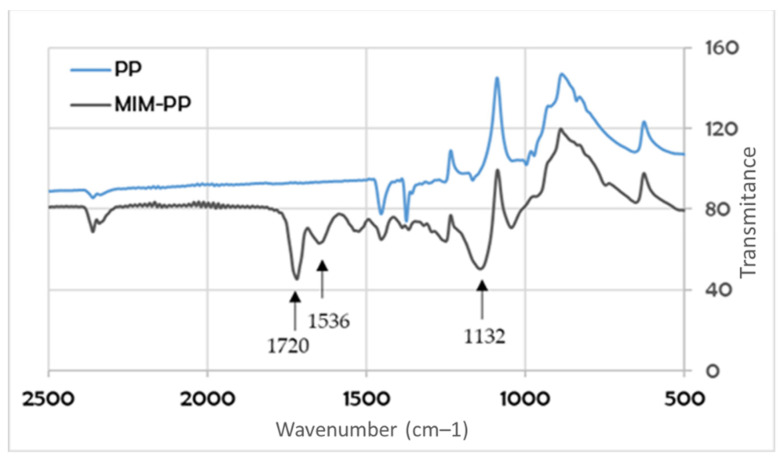
Infrared (IR) spectrum of a non-modified polypropylene (PP) and a modified molecularly imprinted membrane - polypropylene (MIM-PP), G-series.

**Figure 7 membranes-12-00503-f007:**
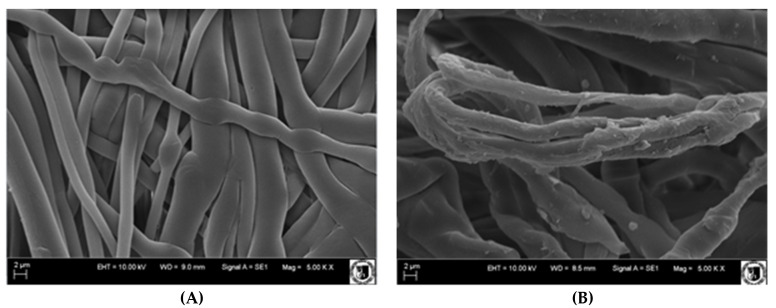
Scanning electron microscope (SEM) images of membranes: (**A**) before modification and (**B**) and with imprinted layer: G-series, the grafting yield (GY) 1.34 mg/cm^2^.

**Table 1 membranes-12-00503-t001:** Composition of the polymerization mixtures.

Series	Monomers Ratio		Monomers Functional: Cross-Linker	Solvent	wt.% DEP
	MMA	NIPAM			
A	1	6	1:1	toluene	5
B				cyclohexanol	5
C				chloroform	5
D				n-octane	5
E					7
F				n-octane/toluene 1:1	5
G	3	7	4:6	n-octane	7
H	4	6	4:6	n-octane	7
I					10

**Table 2 membranes-12-00503-t002:** Sorption properties of imprinted membranes.

Series	MIP		NIP	
	Δm (mg/g)	SS (µmol/g)	Δm (mg/g)	SS (µmol/g)
A	1.098	5.70	0.809	3.86
B	1.002	1.004	1.104	1.12
C	2.659	1.98	2.55	1.66
D	1.792 2.486	1.25 4.23	2.95	3.06
E	1.156 2.584	3.48 5.70	2.58	1.86
F	0.029 3.410 10.57	1.908 3.01 6.78	1.67 3.27	3.35 3.50

**Table 3 membranes-12-00503-t003:** Sorptive properties of membranes modified from solutions with different monomer.

Series	MIP		NIP	
	Δm (mg/g)	SS (µmol/g)	Δm (mg/g)	SS (µmol/g)
G	1.09 1.85	4.90 5.31	1.18 1.91	1.42 1.44
H	0.49 1.67	1.84 4.72	0.47 1.56	1.79 4.07
I	2.197	6.28	2.254	6.43

**Table 4 membranes-12-00503-t004:** Sorption properties of membranes tested in real and model conditions in dynamic sorption.

Membrane	Δm (mg/cm^2^)	*DS* in Model Samples (µmol/g)	*DS* in Real Samples (µmol/g)
imprinted	1.79	4.12	3.09
non-imprinted	1.96	1.18	1.12

**Table 5 membranes-12-00503-t005:** Kinetic parameters of diethyl phthalate (DEP) sorption.

Sample	Pseudo-First		Pseudo-Second	
	*k* _1_	*R* ^2^	*k* _2_	*R^2^*
NIM	0.13	0.978	0.05	0.992
MIM	0.27	0.978	0.33	0.992

**Table 6 membranes-12-00503-t006:** Selectivity results for molecularly imprinted membrane (MIM) and non-imprinted membrane (NIM) membrane.

Parameter	DEP	DBP
K^MIP^	8310	569
K^NIP^	3350	470
α^MIP^	-	14.6
α^NIP^	-	7.1
IF	2.5	1.2
S	-	2.1

**Table 7 membranes-12-00503-t007:** Langmuir and Freundlich and Dubinin–Radushkevich fitting parameters MIM and NIM.

Sample	Langmuir	Freundlich	Dubinin–Radushkevich			
	R^2^	R^2^	*SS_m_* (mg/g)	*SS_m_* (mmol/g)	R^2^	E (kJ/mol)
MIM	0.795	0.769	0.03	1.3 × 10^−4^	0.961	0.07
NIM	0.256	0.234	NA	NA	0.333	NA

NA, not available to calculate.

**Table 8 membranes-12-00503-t008:** Membrane filtration characteristic.

Mass of Grafted Layer Δm (mg/cm^2^)	Water Flux J (L/m^2^ h)	Average Pore Size r (µm)	Rejection Coefficient R (%)
0—non modified	121 ± 1.71	1.29	0.90
0—after plazma treatment	130 ± 2.12	1.30	-
0.026	118 ± 1.41	1.29	8.28
0.503	99 ± 2.06	1.18	12.67
0.712	93 ± 1.14	1.16	17.98
1.261	74 ± 1.35	1.04	26.64
1.544	68 ± 1.56	0.97	28.85
1.916	62 ± 1.34	0.93	36.12
2.088	61 ± 1.48	0.91	35.08
NIM—1.359	-	-	7.92

**Table 9 membranes-12-00503-t009:** The dynamic contact angle and surface energy of the G-series membrane (grafting yield (GY) 1.82 mg/cm^2^).

Membrane	Δm (mg/g)	Contact angle (°)	Surface Energy (mJ/m^2^)
non-modified	0	89	32.4
modified	0.048	68	40.6
	0.353	50	47.0
	1.080	36	52.2

## Data Availability

Not applicable.

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
