# Peer review of "Molecularly Imprinting Microfiltration Membranes Able to Absorb Diethyl Phthalate from Water"

_membranes, 2022, doi:10.3390/membranes12050503_

Round 1
Reviewer 1 Report
In the present manuscript, the authors have studied the Molecularly imprinting microfiltration membranes for polishing water from diethyl phthalate. Overall, the work carried out is interesting and very much relevant to the present-day need. But it requires major revision before accepting the manuscript for publication. The study is confusingly written. Following suggestions must be considered during the revision of the manuscript
It is important to emphasize the main aim of the present study.
Abstract should summarize and contain main results and findings.
The flow of the introduction is not clear and it is not attractive to the readers. The authors must clearly explain the novel explorations made by them.
The methods used in this research should be listed in Introduction part.
What is real sample?
A more detailed explanation of the membrane modification method is needed.
Experimental procedure for DEP sorption should be better explained.
Authors should explain a importance of parameters listed in Table 1.
„A“ is missing in equation 1
What are: S, t and V in equation 2?
Poor discussion on the results. Most of them are general statements. Supporting literature must be cited.
It is also suggested to tabulate and compare results obtained in this study with previously published data related to similar findings.
Author Response
Dear Reviewer,
We thank you for all comments and suggestions to our manuscript. We have considered the raised queries as essential for improvement quality of our work. Please find our answers. New changes in the text of manuscript were introduced in blue.
Best regards
Katarzyna Smolinska-Kempisty

Reviewer 2 Report
The manuscript describes a very interesting topic – the preparation of a molecular imprinted membrane surface for the selective removal of diethyl phthalate from water. The authors used a very open microfiltration membrane to allow a low-pressure operation process but aiming for a high and specific adsorption capacity towards the plasticizer diethyl phthalate (DEP).
The introduction describes the potential health risk that the exposure of DEP poses to humans and justify the necessity of its removal. Then, they report different methods that are in discussion for the removal of DEP but so far, there is no specific or efficient technique present. This could be solved by a molecular imprinted polymer (MIP) membrane surface which should – in the ideal case – adsorb DEP with high selectivity.
The authors aim for modification of a microfiltration membrane (because of low pressure/energy demand) using a plasma-initiated grafting method. What they don’t specifically refer to are previous works that already investigated the preparation of MIP with exact the same monomers and solvents plus monomer ratios for the purpose of either adsorbing bisphenol A or DEP, respectively (19. Wolska, J.; Bryjak, M.; Removal of bisphenol a from aqueous solution by molecularly imprinted polymers. Separation Science and Technology. 2014, 11, 1643-1653. / 29. Wolska, J.; Thermoresponsive molecularly imprinted polymer for rapid sorption and desorption of diethyl phthalate. Separation Science and Technology. 2016, 51, 2547-2553.). In my opinion, this should have been mentioned in the introduction together with the explanation why the present work is innovative compared to the previous publications.
There are several questions open and some typos/errors that I recommend to address before considering the acceptance of this manuscript for publication:
- Title: I think calling the removal of 35% DEP "polishing" is an exaggeration. I would expect higher adsorption rates for that.
- Line 45: please provide definition of abbreviation “DEP”
- Figure 1: this scheme is not very clear, I guess the DEP particles should stick to the imprinted layer and not go through the membrane? Perhaps you could "catch" more of the yellow DEP particles within the layer to make it clear.
- Line 86: please provide definition of abbreviation “PP”
- Line 88: delete “-“ in volt-tage
- Line 97: delete “e” in Soxhlete
- Table 1, head: funcional:cross-linker : what does this mean? Is the crosslinker EGDMA? Is it related to a molar ratio of the monomers and the crosslinker?
- Equation (1): surface area is missing in equation (replace M1)
- Equation (2): what is “S”? There is no definition provided. Shouldn’t this be the area?
- Line 111: Often "r" means pore radius which would be half of the pore diameter ("pore size"). Please be specific.
- Chapter 2.3.4: What is a realistic concentration of DEP in the environment? What was the size of the membrane in this experiment?
- Line 120: space is missing: concentrationwas
- Chapter 2.3.6: What solvents were used? Usually you need the contact angles of two different solvents to determine the surface energy?
- Line 143: typo: measurie
- Figure 2: The IR spectra a not in a condition to interpretate them, it seems that the base line correction has been missing? It is not clear if this is measured in transmission or absorbance mode (y-axis is missing). Please correct these errors!
- Lines 164/165/167: cm-1 is not a wavelength but a wavenumber!!
- Line 167: space missing: 1132cm-1
- Line 169: you are referring to Figure 2 but there is no EGDMA moleucule in Figure 2?
- Table 2: How was the surface energy calculated when contact angles were only determined with water? Normally, at least two different solvents are required to calculate the surface energy. Then, also the ratio of polar and dispers contribution to the surface energy should be provided, otherwise the data doesn't provide more information than the contact angle already does.
- Table 2: Which of the modified membranes according to table 1 was presented here? Why there are no information of the different modifications according to table 1 and their respective grafting degrees and contact angles?
- Line 187: You can't calculate the pore size of a membrane when roughness and hydrophilicity are changing as well! I don't understand the equation that you used here. Your membrane has no tubular-shaped pores but is a kind of nonwoven. When you measure a flux change of less than 10% of your pristine membrane I anyway would not call this very significant unless you also provide error ranges...
- Line 194: I don't agree with this assumption. You said plasma treatment causes etching, furthermore, you graft a new layer on top of your membrane. In the SEM pictures you see a kind of microstructure appearing (roughness changes) plus you change the wettability/hydrophilicity of your membrane surface (you didn't provide the contact angle after plasma treatment!). It is well known that plasma treatment leads to a temporary increase in surface wettability which can also explain the change in permeability. Please also provide the thickness of the membrane if you discuss the fact the plasma was able to modify also the cross-section of your membrane. I would also request to provide XPS data of the top and the bottom of the membrane to confirm this assumption here!
- Line 198: Why the pore size is no changed? In your calculation of the pore size the only parameter that changes is the flux, so why the pore size is not changing in the first case but with increasing grafting yields you see an effect?
- Line 207: "by" not "to"
- Table 3: Water flux: please provide error ranges!
- Table 3: How does this correlate with table 1?
- Figure 4: Please provide also the rejection coefficient of non-imprinted membrane
- Line 214: Is it particles or molecules? if particles: please provide particle size!
- Line 217: grammar: “has” not “have”
- Line 228: If you compare MIM with NIM regarding the adsorption of a similar target, you are discission the specific adsorption of the membrane but not selectivity. I wonder why the authors use a wrong technical term after they already published several works on MIP polymers…Nevertheless, it would be very interesting to also investigate the selectivity of the membranes towards different targets! However, since it seems that the authors already investigated the same systems towards bisphenol A adsorption, they could at least compare the results at this point here…
- Line 234: please remove “very”. Either your material is nonspecific or it is not. But it can’t be VERY nonspecific.
- Line 239: start sentence with “When increasing the number of the template…”
- Line 239: replace “number” by “concentration”
- Line 246: add “to” 4:6
- General comments on results chapter:
Please also check if you can regenerate the membranes and how often? What solvents can you use for this purpose? Is the MIM Membrane stable enough for the conditions?
Please discuss the properties of your material with already known materials (e.g. 19. Wolska, J.; Bryjak, M.; Removal of bisphenol a from aqueous solution by molecularly imprinted polymers. Separation Science and Technology. 2014, 11, 1643-1653. / 29. Wolska, J.; Thermoresponsive molecularly imprinted polymer for rapid sorption and desorption of diethyl phthalate. Separation Science and Technology. 2016, 51, 2547-2553.) In the Paper 19, the authors describe a similar membrane for removal of bisphenol A. This should be discussed within the results obtained within this work since that would mean that the membrane is not selective towards DEP if it also adsorbs bisphenol A in large amounts...IN the Paper 29, NIPAM:MMA with a ratio of 3:7 and toluene as a solvent have been identified as the best system to adsorb DEP, how does this fit to the results of the present work?
- Line 255: replace “removed compound” by DEP
- Line 255: add: Thanks to this it "is" possible
- Line 256: please provide definition for abbreviation “ED”
- Line 261: please correct: 7 mol-%
- Line 264: please delete: “the” average.
- Line 264: please add punctuation mark at the end of the sentence.
- Line 265: please replace “on” by “of”.
- Line 266: this is wrong, please replace “5” by “4”
- Line 268: Where does this number (64%) originate from? I don't find it in the results discussion...
- Page 8: What is the reason for octane being the best solvent? Is there any idea? Is it swelling? Are there more open cavities formed to be accessed by the template? Is the template better soluble in octane compared to the other solvents? Has the overal surface area been investigated to check if there are more cavities for the template? I guess the authors perhaps know more about the mechanism behind since they already published some works about this polymer system as a MIP for adsorbing DEP and bisphenol A…
Author Response
Dear Reviewer,
We thank you for all comments and suggestions to our manuscript. We have considered the raised queries as essential for improvement quality of our work. Please find our answers. New changes in the manuscript text were introduced in blue.
Best regards
Katarzyna Smolinska-Kmepisty

Reviewer 3 Report
The manuscript describes the appplication Molecularly imprinting microfiltration membranes for removal of diethyl phthalate
I think the term polish in the title is not appropriate and should rather be rewritten and removal or treatment of DEP, The Language of the text should also be revised and corrected for many grammatical mistakes.
No information were given in the text regarding the adsorption properties or the kinetics of the designed membranes
The imprinting efficiency is not given and no information was also given regarding the removal of the template from the imprinted membranes prior to their application for different optimization parameters.
No information was given regarding the selectivity of the modified membranes
no information was given regarding the reusability or regeneration of the membranes
the application section is poor and data should be compared to a standard reference method.
The efficiency of the proposed membranes should be compared to bare membranes and other previously reported methods for modification of pp membranes
Author Response

(The authors gave the same response as above.)

Round 2
Reviewer 1 Report
The authors corrected Manuscript according to reviewer suggestions.
Author Response
Dear Reviewer,
we hope this version of the manuscript is more correct.
Yours faithfully
Katarzyna

Reviewer 3 Report
Still the language needs improvement as in many parts it was really hard to understand what was the authors approach. and the application section is still poor
For this question, data for kinetics were given and nothing about the nature of interactions between sorbents and samples which can be revealed not don't by kinetics but also by testing what kind of isotherm.
For the selectivity section, one interferent can never be enough at all, especially multiple possible others can be there
For reusability or regeneration of the membranes section, its poorly written and no error bars were given for the presented bar chart
Still the application section is poor, and as indicated it is just a pilot study!!!
Author Response

(The authors gave the same response as above.)
